# T Cell Response to SARS-CoV-2 Coinfection and Comorbidities

**DOI:** 10.3390/pathogens12020321

**Published:** 2023-02-15

**Authors:** Liqing Wang, Hao-Yun Peng, Aspen Pham, Eber Villazana, Darby J. Ballard, Jugal Kishore Das, Anil Kumar, Xiaofang Xiong, Jianxun Song

**Affiliations:** 1Department of Microbial Pathogenesis and Immunology, Texas A&M University Health Science Center, Bryan, TX 77807, USA; 2Department of Biochemistry and Biophysics, Texas A&M University, College Station, TX 77843, USA

**Keywords:** SARS-CoV-2, T cells, coinfection, comorbidities

## Abstract

For the past three years, COVID-19 has become an increasing global health issue. Adaptive immune cells, especially T cells, have been extensively investigated in regard to SARS-CoV-2 infection. However, human health and T cell responses are also impacted by many other pathogens and chronic diseases. We have summarized T cell performance during SARS-CoV-2 coinfection with other viruses, bacteria, and parasites. Furthermore, we distinguished if those altered T cell statuses under coinfection would affect their clinical outcomes, such as symptom severity and hospitalization demand. T cell alteration in diabetes, asthma, and hypertension patients with SARS-CoV-2 infection was also investigated in our study. We have summarized whether changes in T cell response influence the clinical outcome during comorbidities.

## 1. Introduction

Since its start at the end of 2019, the COVID-19 outbreak has posed significant health risks to the human population, surpassing more than 6.5 million deaths across the world (WHO cumulative data up to Oct 2022). Over the past 3 years, publications regarding coinfection with SARS-CoV-2 and other pathogens, such as viruses, bacteria, and parasites, have been reported. Tissue damage resulting from SARS-CoV-2 includes lung failure [1], brain damage [2], intestinal mucosal damage [3], and cardiovascular disease [4]. Furthermore, in cases when a cytokine storm is present in SARS-CoV-2 infection, coinfection may exacerbate the outcomes of other types of diseases, such as allergy, diabetes, and hypertension.

T cells play a pivotal role during viral infection and immune-related diseases. Due to SARS-CoV-2’s ability to alter the T cell response, it may affect the outcomes of several common pathogenic infections and chronic diseases. In a reported case of HIV coinfection with SARS-CoV-2, depletion of CD4^+^ T cells was observed to affect the patient’s clinical outcome [5]. In other conditions such as TB infection and diabetes, there is an increased risk of severe SARS-CoV-2 due to exacerbation of symptoms [6]. In diabetic patients, the increased clinical risk is positively correlated with SARS-CoV-2 infection [7].

In this review, we briefly summarize the infection route of SARS-CoV-2 and the different T cell subsets’ responses during SARS-CoV-2 infection. We highlight T cell performance during SARS-CoV-2 coinfection with HBV, HCV, HIV, influenza, TB, and parasites. We also emphasize the T cell response in chronic patients with allergy, diabetes, and hypertension during SARS-CoV-2 infection. 

## 2. SARS-CoV-2 Infection and T Cell Response

Within lethal contagious respiratory viruses, the most prevalent transmission route is an airborne infection [8]. In Figure 1, when the SARS-CoV-2 reaches the lung’s epithelial layer, the spike protein on the viral surface will bind with the ACE2 receptor, accelerating the entry of the virus [9,10]. Fusion of the virus and host cell membrane results in the release of the viral genome [11]. Essential components (including N, S, M, and E proteins) and sub-genomic viral RNA are subsequently generated and assembled to form an intact virus [12]. The newly generated virus is released from the cytoplasm through exocytosis [12] and SARS-CoV-2 will be captured by antigen-presenting cells (APC). Viral antigens are then processed and presented to the T cells [13,14]. Meanwhile, the defense mechanism mediated by T cells with SARS-CoV-2 is different from the antibody-dependent response due to their ability to recognize a broad range of viral epitopes [13,15]. Compared with the common cold, SARS-CoV-2 infection generates a diverse epitope pool and increases the frequency of both viral-epitope-specific CD4^+^ and CD8^+^ T cells among convalescent SARS-CoV-2 patients [13]. In terms of infection, T cells secrete IFNγ, Granzyme B, and TNFα to help eliminate SARS-CoV-2 infected cells [16]. In mild COVID-19 patients, the effector CD4^+^ or CD8^+^ T cells will proliferate and form a defense mechanism during the acute phase [17]. In moderate-symptom patients, the natural killer T cell (NKT) CD160 population responds quickly by direct cytotoxicity [18]. Furthermore, the SARS-CoV-2-specific memory T cells are beneficial for providing essential protection against future reinfection [15,19,20,21]. 

## 3. T Cell Subsets and Surface Markers Change during SARS-CoV-2 Infection

As studied, SARS-CoV-2 patients with severe symptoms often displayed progressive lymphopenia, while patients with mild symptoms showed normal absolute lymphocyte counts [22]. Within the CD8^+^ T cell populations in these patients, an increased population of CD8^+^ effector memory T cells was observed [22]. On the other hand, lower CD4^+^ or CD8^+^ frequencies were detected in severe COVID-19 patients compared to mild COVID-19 patients [23]. A study performed by Neidleman et al. indicated that CD27^+^CD28^+^CD8^+^ T effector memory cells re-expressing CD45RA (T_EMRA_) were predominant in the recovery period of SARS-CoV-2 patients, contradicting the expression of the CD27 marker [24]. Moreover, there was an increase in both central memory CD8^+^ T cells (CD45RA^−^CD27^+^CCR7^+^) and effector memory CD8^+^ T cells (CD45RA^−^CD27^+^CCR7^−^) [25,26,27]. In regards to Ki67^+^ and HLA-DR markers co-expressed on CD8^+^ T cells, SARS-CoV-2 infection led to an increase in Ki67^+^HLA-DR^+^CD8^+^ T cells in the patients [26]. In addition, there was an increase in the expression of activation markers, including HLA-DR and CD45RO, on CD8 T cells in severe SARS-CoV-2 patients compared to mild patients [28]. An increased proportion of CD38^hi+^ CD8^+^ T cells was also observed in severe SARS-CoV-2 infected patients [26,29,30]. In addition, a reduction in CD27^+^CD8^+^ T cells and an increase in CD127^+^CD8^+^ T cells were found in SARS-CoV-2 patients [24]. When compared to healthy blood donors, CTLA-4, LAG-3, and Tim-3 were significantly expressed in memory CD8^+^ T cells from patients with severe SARS-CoV-2 infection, while there was no difference for the inhibitory immune checkpoint PD-1 marker [26,30]. 

Cords detected that LAG-3 and TIGIT expression were upregulated in SARS-CoV-2-specific CD4^+^ T cells when compared to CD4^+^ T cells in healthy patients [31]. A higher percentage of cells expressing HLA-DR and CD45RO were also observed within the effector CD4^+^ T cell population in severe patients compared to mild patients [32]. On the other hand, a reduction in CD28^+^CD4^+^ T cells was detected in severe patients compared to mild patients [32]. Cytokines, including IL-2R, IL-6, IL-8, and IL-10, experienced an increase in production from effector CD4^+^ T cells in severe SARS-CoV-2 patients [33,34]. The decreased expression of CD45RA and CCR7 indicated that T central memory cells are more dominant than T effector memory cells [24]. It was also reported that there were high populations of memory CD4^+^ T cells expressing CD38, CD69, Ki-67, and PD-1 in memory CD4^+^ T cells from patients with severe SARS-CoV-2 infection compared to healthy blood donors [26].

Among CD4^+^ T cell subsets, CD4^+^ Teff cells differentiate into functionally diverse subsets such as Th1, Th2, Tfh, Th17, and Treg cells. T helper 1 (Th1) cells protect against intracellular parasites by the secretion of IFNγ [35,36]. Studies have indicated that Th1 cells predominate during SARS-CoV-2 infection through upregulation of IFNγ but not IL-4 or IL-17. T helper 2 (Th2) cells help to eliminate extracellular pathogens [36]. Studies also indicate that Th2 and Th17 cells play a role in SARS-CoV-2 infection in which Th2 cells were claimed to be unfavorable for recovery [24,30,37]. Lower levels of Th2 populations were observed in mild patients compared to healthy donors, but no difference was indicated between mild and severe patients. The disproportionate ratio of Th2/Th1 cells revealed a lead in severe infection, with abnormal secretion of associated cytokines such as IL-4, IL-5, IL-10, and IL-13 [30,38,39,40]. T follicular helper cells (Tfh), a particular subset of CD4^+^ T cells that regulates antibody responses with B cells, have been noted to play a role in SARS-CoV-2 infection [22]. Mathew et al. indicated that there was a decreased population of CD45RA^−^CXCR5^+^PD-1^+^ circulating Tfh (cTfh) cells post-SARS-CoV-2 infection with an upward trend of Tfh cell population in recovered patients within six months [27]. Among cTfh cells, there are multiple subsets including Tfh1, Tfh2, and Tfh17 that are categorized based on differences in the secretion of cytokines. A lower population of cTfh17 (CXCR5^+^CXCR3^−^CCR6^+^) cells, which secrete IL-21 and IL-17A, was discovered in a common variable immunodeficiency disease (CVID) patient with SARS-CoV-2 infection compared to the immunocompetent donor [22,41]. However, for PD-1 marker expression, there were multiple contradictory studies indicating that there were no changes in PD-1^+^ Tfh cells in SARS-CoV-2 patients [42]. A high expression of ICOS was also found in SARS-CoV-2-specific Tfh cells [24]. In regard to Treg cells, which act as suppressors in the immune system, there are conflicting studies. An increased percentage of regulatory T cells (CD3^+^CD4^+^CD25^+^CD127^low^) and a higher secretion of IL-10, TGF-b, IL-6, and IL-18 were detected during COVID-19 infection, specifically with N peptide stimulation [43,44]. Increased populations of Treg cells were present in mildly symptomatic patients compared to healthy controls [43]. As a result of these observations, Treg cells may potentially play a role in assisting with SARS-CoV-2 recovery. However, some conflicting studies reported that a reduction in Treg cells and lower expressions of FOXP3, TGF-β, and IL-10 were present in patients with severe SARS-CoV-2 infection [45,46,47]. Therefore, more research is required for the role of Treg cells. A high population of Th17 cells, which act as inflammatory cells in autoimmune diseases, was found in SARS-CoV-2 patients, with the detection through the expression levels of Th17 cells’ signature transcription factor RAR-related orphan receptor gamma [RORγt] and the secretion levels of signature cytokines such as IL-6, IL-17, and IL-23 [24,48]. Changes in T cell subsets and surface markers during SARS-CoV-2 infection are summarized in Figure 2.

## 4. Role of T Cells during Coinfection with Viruses, Bacteria, and Parasites

During the three-year pandemic, researchers have noticed cases of coinfection with SARS-CoV-2 and other viruses (e.g., HBV, HIV, HCV, and influenza), bacteria (e.g., Mtb), and parasites (e.g., protozoa and helminths) (Figure 3).

### 4.1. T Cells’ Role during SARS-CoV-2 Coinfection with HBV, HIV, HCV, and Influenza

Hepatitis B virus (HBV) is a contagious virus that causes chronic liver diseases such as fibrosis and hepatocellular carcinoma (HCC) [49]. Currently, the coinfection outcome of SARS-CoV-2 and HBV is still ambiguous. A report displayed that 3% of SARS-CoV-2 patients will suffer from chronic liver disease [50]. Some researchers reported cases where coinfection resulted in severe liver injury and other severe outcomes [51,52,53]. Clinical studies show that coinfected patients experience a high severity and poor prognosis, both of which should be taken seriously [54]. However, other publications indicate that no significant difference has been observed in the clinical outcome of chronic HBV carriers who were SARS-CoV-2 coinfected [55,56]. On one hand, T cells are important for a host to defend against the invasion of SARS viruses. On the other hand, it is well known that SARS-CoV-2 will induce excessive cytokine storms from T cells, causing increased severity [57]. In terms of immune suppression by HBV infection, T cell exhaustion is accelerated, and partial cytokine secreted T or NK cells become dysfunctional [58,59]. These counterparts of coinfection may explain the lack of significant difference observed for HBV patients coinfected with SARS-CoV-2 [60]. 

Human immunodeficiency virus (HIV) infection has been confirmed by multiple studies to attack CD4^+^ helper T cells and destroy immune system functionality [61]. In addition, HIV infection also causes T cell exhaustion [62]. These studies suggest that HIV destroys and deteriorates the immune system more severely than HBV, although both viruses will cause T cell exhaustion. As stated in a UK case report, coinfection of SARS-CoV-2 and HIV increases the risk of death [63]. In addition, a study carried out in New York indicated that the coinfected population experienced deleterious outcomes compared with the general SARS-CoV-2 infection [5]. Another research study reported that HIV patients treated with proper antiretroviral therapy (ART) were not generally affected by SARS-CoV-2 infection. However, without ART treatment, the comorbidity of these patients increased severely [64]. 

Similar to HBV, Hepatitis C virus (HCV) is also a harmful liver virus, causing more than 400,000 deaths annually [65]. In HCV patients with liver cirrhosis, the severity and mortality during coinfection with SARS-CoV-2 were increased [66]. A case report indicates that the SARS-CoV-2 RNA shedding is prolonged in the coinfection [67]. However, due to limited data reported for HCV and SARS-CoV-2 coinfection, T cell involvement in these cases is still under ambiguous investigation.

Influenza infection shares a similar clinical presentation compared to SARS-CoV-2. For example, both infections trigger symptoms of fever, headache, and respiratory tract inflammation [68]. Over the past several decades, T cells’ response to influenza viral infection has been investigated [69]. It has been shown that both CD8^+^ and CD4^+^ T cells respond to the influenza virus (IAV) and provide protective mechanisms during infection [69,70]. Several case reports and analyses have been published regarding the coinfection of influenza and SARS-CoV-2 [71,72]. These reports indicate that coinfection leads to a higher risk of death and severity compared to a single infection [68,71] due to IAV’s ability to induce the ACE2 receptor expression [68]. Furthermore, coinfections of SARS-CoV-2 and IAV seem to impair CD4^+^ T cell response and decrease neutralizing antibody efficacy [73]. Researchers have also discovered that prior IAV immunity may ameliorate coinfection due to IAV-antibody-dependent immunity studied in mice experiments [74]. In a hamster model, researchers noted that following IAV pre-infection, SARS-CoV-2 replication was interrupted [75]. Therefore, it has been concluded that the influenza vaccine may be beneficial for individuals seeking to decrease the risk of SARS-CoV-2 infection. 

### 4.2. T Cells’ Role during SARS-CoV-2 Coinfection with Mycobacterium tuberculosis (Mtb)

*Mycobacterium tuberculosis* (Mtb) is an aerobic bacterium that attacks human lung cells [76]. Although the Bacille Calmette–Guérin (BCG) vaccine is available and provides sufficient protection for infants and children, adults are still vulnerable to infection [77]. Mtb infection affects the immune system by causing delays in the initial activation of both Mtb-specific CD4^+^ and CD8^+^ T cells [78]. In a reported study, one article suggests that CD4^+^ T cells may be capable of preventing prevent CD8^+^ T cell exhaustion during Mtb infection [79]. Since both Mtb and SARS-CoV-2 infections cause damage to lung tissue, coinfection has been associated with a higher death rate compared with SARS-CoV-2 infection only [6]. In coinfection, researchers have discovered that SARS-CoV-2 patients display a decreased frequency of Mtb-specific CD4^+^ T cells [80]. Furthermore, SARS-CoV-2 patients infected with either TB or latent TB showed a lower response to SARS-CoV-2 and had a decreased Mtb-specific immune response [81]. 

### 4.3. T Cells’ Role during SARS-CoV-2 Coinfection with Parasite

Parasite infection is a long-standing area of concern in the field of public health. In response to common parasitic infections, such as protozoa and helminths, the body will induce regulatory T cells to suppress the host’s antiparasitic immune system [82]. Multiple researchers have discovered that intestinal parasite coinfection with SARS-CoV-2 decreases SARS-CoV-2-mediated intestinal inflammation risk [83]. In addition, parasite-driven Th2 and Treg cell responses were able to counterbalance SARS-CoV-2-induced cytokine storms [83,84]. This discovery may explain why parasite coinfection with SARS-CoV-2 may ameliorate the severity of SARS-CoV-2 patients.

## 5. T Cells’ Response to Comorbidities—Diabetes Mellitus, Asthma, and Hypertension

It has been observed that severe cases of SARS-CoV-2 infection were experienced by elderly or comorbid patients, including those with diabetes mellitus, asthma, and hypertension. However, the pathogenic mechanisms and how T cells respond are still not well-understood. Understanding the T cell response to several comorbidities may lead us to understand the improvement of health span (Figure 4). 

### 5.1. T Cells’ Response to COVID-19 Comorbidities—Diabetes

Diabetes, characterized by hyperglycemia, has been a tremendous health issue globally. There are two major types of diabetes: type 1 diabetes (T1D) is distinguished by insufficient levels of insulin while type 2 diabetes (T2D) is characterized by the ineffective use of insulin by cells [85]. It has been reported that there is an increase in inflammation along with a downregulation of immune response in diabetes patients. Furthermore, some studies suggested that diabetes may drive the severity of SARS-CoV-2 infection upward [86,87,88]. Prior to SARS-CoV-2 infection, Wang et al. observed that diabetic patients with glucose tolerance dysfunction have a 50% risk of pneumococcal infection compared to healthy individuals [87,88]. Therefore, diabetic patients with SARS-CoV-2 infection are considered to have a higher risk of disease severity and death [88]. In diabetic patients, a lower CD4^+^ T cell population was reported whereas in diabetic patients infected by SARS-CoV-2, a higher percentage of CD4^+^ T cells and a lower percentage of CD8^+^ T cells were detected [86,89]. Furthermore, an increase in cytokine production of IL-2, IL-6, IL-8, IL-10, TNFα, and IFNγ was observed in coinfected patients [89,90]. A study by Han et al. indicated that a lower Th1/Th2 cytokine ratio was observed in diabetic patients when compared to nondiabetic patients [90]. 

### 5.2. T Cells’ Response to COVID-19 Comorbidities—Asthma

Asthmatic patients who were exposed to certain allergens such as chemical irritants, dust mites, grass pollen, or animal dander experienced inflammation, leading to narrowing of the airway. Common symptoms of asthma are cough, shortness of breath, and chest tightness. Initially, researchers thought that respiratory illnesses such as allergy and asthma caused potential risks for SARS-CoV-2 patients primarily due to excessive mucus production and damaged epithelium [91,92]. Moreover, a higher expression of TMPRSS2, a transmembrane host protease that is essential for viruses to enter the host, was reported in asthmatic patients [48]. In an analysis platform study, 23 million medical records from severe SARS-CoV-2 patients noted that severe asthma was associated with increased risk [91]. It has been known that asthma is a CD4-mediated disease, specifically affecting Th2 cells. When patients inhale certain aeroallergens, allergic-specific Th2 cells are activated, causing inflammation and chest tightness. Typically, there is a higher risk of viral infection, such as SARS-CoV-2, associated with high Th2 populations [93]. Another study suggested that an imbalance between Th2 and Th1 cells may increase the mortality rate in SARS-CoV-2-infected asthmatic patients [40]. Furthermore, Pavel et al. noted that inhibition of Th2 cells may provide protection from SARS-CoV-2 infection [40]. This discovery has led to studies showing that allergic asthma decreased the hospitalization rate and produced no effect on severe SARS-CoV-2 patients [94,95]. On the other side, there is a conflicting study showed that the recovery rate for SARS-CoV-2 patients with asthma and without asthma is similar, possibly due to the low expression of ACE2 in asthmatic patients [94,96]. Asthmatic patients with no controller treatment have lower expression of ACE2 compared to asthmatic patients with treatment. However, there are conflicting results discussing how asthma therapy may reduce the severity of SARS-CoV-2 infection [94,97]. In a study by Matsuyama et al., the inhaled steroid ciclesonide was shown to inhibit SARS-CoV-2 RNA replication, while another study stated that there is no association between asthma medications and SARS-CoV-2 clinical outcomes [94,97]. In order to have a better understanding of the T cell response to asthma–COVID-19 comorbidity, further investigation is needed. 

### 5.3. T Cells’ Response to COVID-19 Comorbidities—Hypertension

Patients with SARS-CoV-2 and hypertension are more likely to experience elevated blood pressure, develop severe pneumonia, and have increased mortality rates up to double those without hypertension [98,99]. A research study has reported that patients with hypertension generally have higher expression of ACE2 [48]. SARS-CoV-2 patients with hypertension who take antihypertensive treatment were observed to have a lower risk of mortality compared to patients without treatment [99]. When comparing different types of anti-hypertensive treatments, no difference in mortality rate and blood pressure was observed between the renin–angiotensin–aldosterone system (RAAS) inhibitors and non-RAAS inhibitors. Therefore, it is suggested that SARS-CoV-2 patients with hypertension should avoid discontinuing anti-hypertensive treatment [99]. 

Hypertension increases the T cell populations and inflammatory cytokine secretion of tumor necrosis factor TNFα, IFNγ, and IL-17A [100,101]. However, one study indicates that there was no difference in T cell populations between hypertensive patients and healthy people [47]. T cells in hypertensive patients displayed senescence phenotypes with lowered co-expression of CD27 and CD28, resulting in difficulty protecting against viral infections [102]. In addition, some studies indicate that CD4^+^ T lymphopenia was prevalent in patients with SARS-CoV-2 infection and hypertension [103]. The proportion of CD8^+^ T cells that express the activation markers CD38 and HLA-DR was reportedly higher when the hypertensive patients contracted SARS-CoV-2 infection [103] and surprisingly, CD38^+^HLA-DR^+^ CD8^+^ T cells gradually decreased when hypertensive patients recovered from SARS-CoV-2 infection [103]. A high expression of the exhaustion marker PD-1 was also reported on CD8^+^ T cells in hypertensive patients with SARS-CoV-2 infection [103]. In CD4^+^ and CD8^+^ T cells, IFNγ secretion was higher in SARS-CoV-2 patients with hypertension. 

## 6. Conclusions

In this review, we summarized the infection route and replication process of SARS-CoV-2. During infection, T cells are involved in the defensive process and secrete cytokines to defend against invading pathogens. T cell subsets adapt by altering the expression of surface markers or the secretion of cytokines to respond to environmental changes caused by other viral infections and other comorbidities. Subsequently, surface marker expression levels among T cell subsets change during SARS-CoV-2 infection and post-infection.

We summarized SARS-CoV-2 coinfection with pathogens including other viruses, bacteria, and parasites. The outcomes of coinfection are variable depending on different viral species’ influence on T cell response signals. Some viral infections result in latency and T cell exhaustion while virulent viruses deteriorate the T cell immune system and create a more vulnerable host environment for coinfecting. For example, Mtb and SARS-CoV-2 coinfection results in elevated mortality rates due to increased effects on the lungs. On the other side, parasitic infection may work to ameliorate SARS-CoV2-mediated inflammation due to the elaborate effect on T cells. Whether coinfecting pathogens could deteriorate or ameliorate a patient’s condition largely depends on the specific pathogen. 

Other than SARS-CoV-2 coinfections, we summarized how T cells respond to comorbidities such as asthma, diabetes, and hypertension. There are contradictory findings regarding how T cells respond to asthma and SARS-CoV-2 coinfection, due to the higher Th2 populations and lower ACE2 expressions. Diabetes may increase the severity of SARS-CoV-2 infection because of lower levels of CD8^+^ T cells and higher levels of CD4^+^ T cells, specifically an imbalance ratio of Th1/Th2 cells. Lastly, hypertensive patients following SARS-CoV-2 infection may have an increased mortality rate due to higher ACE2 expression, increased levels of inflammatory cytokines, and a T cell senescence phenotype.

## Figures and Tables

**Figure 1 pathogens-12-00321-f001:**
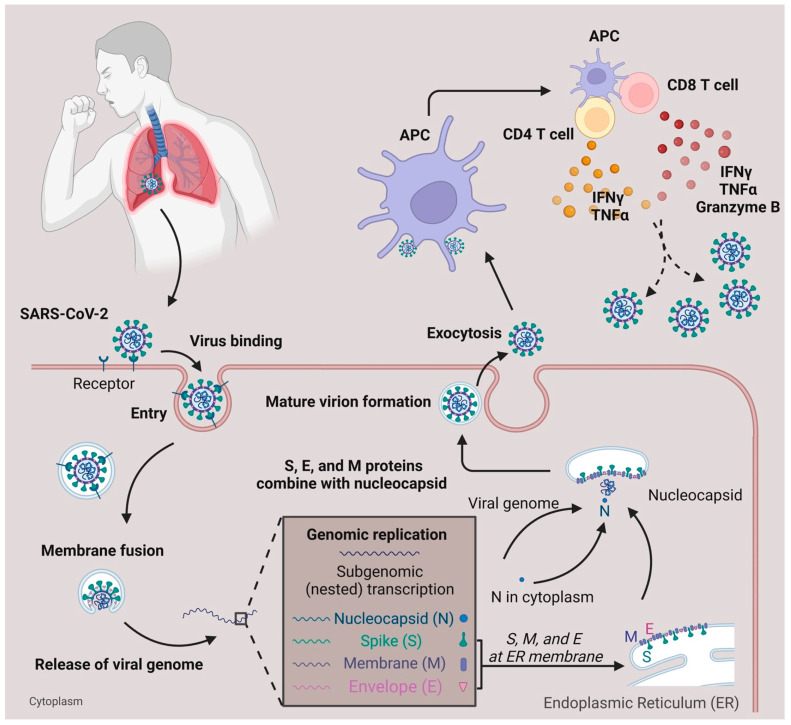
SARS-CoV-2 infection and T cell response. SARS-CoV-2 infects the respiratory tract by binding to the ACE2 surface receptor. Following entry via membrane fusion, it releases its viral genome in the cytoplasm. Using the host cell machinery, viral genome replication, and subgenomic transcription generate essential components needed for the virus to pack together. After the maturation of the virion, it is released from the host cell through exocytosis. When the viruses are captured and processed by antigen-presenting cells (APC), the T cells, either CD4^+^ or CD8^+^, are activated and secrete cytokines (IFNγ, TNFα, Granzyme B) to defend against viruses.

**Figure 2 pathogens-12-00321-f002:**
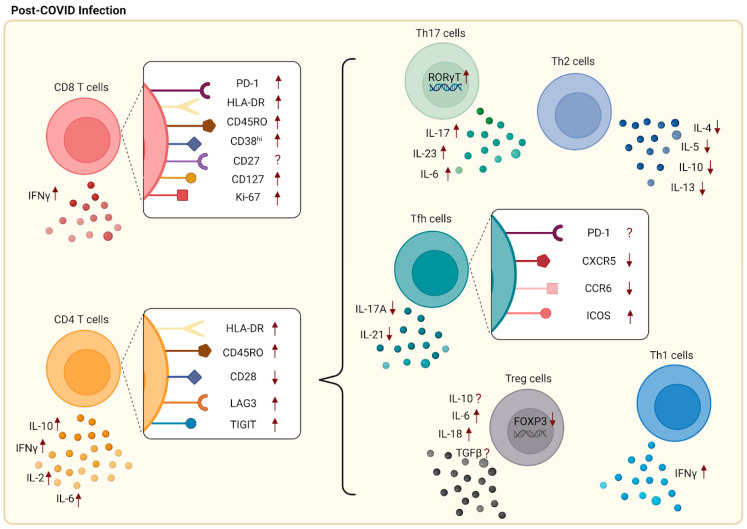
Changes in surface markers and cytokines following SARS-CoV-2 infection. Increased expression of surface markers including PD-1, Ki-67, HLA-DR, CD45RO, CD38hi, and CD127 was found on CD8^+^ T cells, while CD27 shows a contradicting result. Upregulation of LAG-3, TIGIT, HLA-DR, and CD45RO as well as downregulation of CD28 were detected on CD4^+^ T cells. Included in CD4 T cell subsets are Th1, Th2, Th17, Tfh, and Treg cells. Increased populations of Th1 cells along with the production of IFNγ predominated following SARS-CoV-2 infection. Conversely, low numbers of Th2 cells along with abnormal secretions of associated cytokines were found in mild patients. As for Tfh cells, lower expressions of CXCR5 and CCR6 and higher expression of ICOS were observed, while PD-1 expression had conflicting results. There were contradicting studies for Treg cells with some suggesting that high populations of Treg cells were associated with increased secretion of IL-10, TGF-β, IL-6, and IL-18. On the other hand, some researchers indicated a reduction in Treg cells, with lower expression of FOXP3, TGF-β, and IL-10. Lastly, an increased percentage of Th17 cells with high expression levels of RORγt and increased secretion of signature cytokines are found post-SARS-CoV-2 infection.

**Figure 3 pathogens-12-00321-f003:**
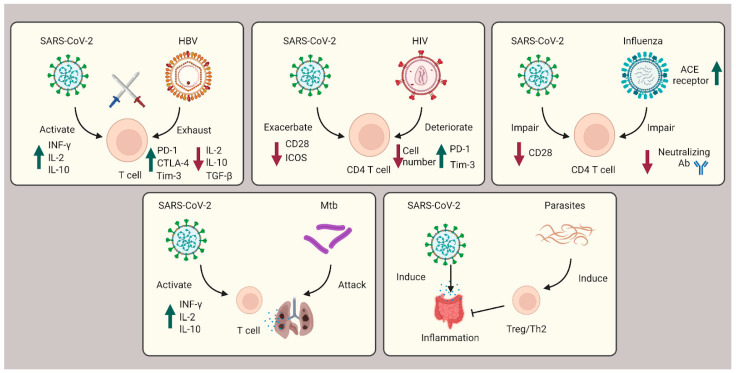
Role of T cells during coinfection with viruses, bacteria, and parasites. In SARS-CoV-2 coinfection with HBV, exhausted T cells are caused by HBV infection and subsequently countered with SARS-CoV-2 infection cytokine storms. In HIV coinfection, CD4^+^ T cells are corrupted, leading to exacerbated patient outcomes. Influenza coinfection decreases neutralization antibody efficacy and increases ACE2 receptors, overall boosting SARS-CoV-2 infection. For Mtb coinfection, both infections will affect the lung tissue, resulting in respiratory failure. In coinfection with parasites, Th2 and Treg cells are induced to suppress the immune system and ameliorate the intestinal inflammation severity of SARS-CoV-2.

**Figure 4 pathogens-12-00321-f004:**
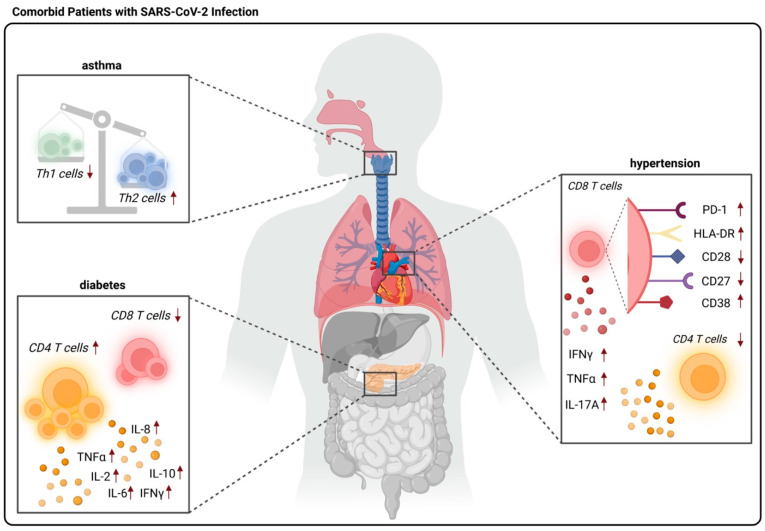
T cells’ response in comorbid patients with SARS-CoV-2 infection. There is a disproportionate ratio of Th1/Th2 cells as well as lower populations of Th2 cells in asthmatic patients following SARS-CoV-2 infection. CD4 T lymphopenia is observed in hypertensive patients with SARS-CoV-2. Surface markers such as PD-1, HLA-DR, and CD38 are highly expressed while CD27 and CD28 are lightly expressed in CD8 T cells. In addition, there is a reduction in CD8 T cells and an elevation in CD4 T cells with cytokines IL-2, IL-6, IL-8, IL-10, IFNγ, and TGFβ in diabetic patients with SARS-CoV-2 infection.

## Data Availability

Not applicable.

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
