# Peer review of "T Cell Response to SARS-CoV-2 Coinfection and Comorbidities"

_pathogens, 2023, doi:10.3390/pathogens12020321_

Round 1
Reviewer 1 Report
This is a well written manuscript which describes in detail the T-cell response to SARS-CoV-2 coinfection with other pathogens, and comorbidities such as asthma, diabetes, and hypertension. These comorbidities are predisposing conditions for poor outcomes in COVID-19 patients. Future research on the role of the T-cells response to COVID-19 comorbidities may well help shed light on the various manifestations of the disease. I think that the manuscript provides a great deal of useful knowledge on the topic and it could be published in the current form.This is a well written manuscript which describes in detail the T-cell response to SARS-CoV-2 coinfection with other pathogens, and comorbidities such as asthma, diabetes, and hypertension. These comorbidities are predisposing conditions for poor outcomes in COVID-19 patients. Future research on the role of the T-cells response to COVID-19 comorbidities may well help shed light on the various manifestations of the disease. I think that the manuscript provides a great deal of useful knowledge on the topic and it could be published in the current form.This is a well written manuscript which describes in detail the T-cell response to SARS-CoV-2 coinfection with other pathogens, and comorbidities such as asthma, diabetes, and hypertension. These comorbidities are predisposing conditions for poor outcomes in COVID-19 patients. Future research on the role of the T-cells response to COVID-19 comorbidities may well help shed light on the various manifestations of the disease. I think that the manuscript provides a great deal of useful knowledge on the topic and it could be published in the current form.
Reviewer 2 Report
Wang L, Peng HY et al attempt to summarize the current understanding of T cell responses in SARS-CoV2 co-infection with other pathogens and comorbidities. Given the increase in severity and mortality in these cases, it is pertinent to understand the underlying T cell response and its effect on clinical outcome. This is a timely report trying to parse the slew of conflicting reports published during the last three years of the pandemic.
Major concerns:
· The wrong figure has been assigned in place of Figure 2.
· In the section delineating T cell subsets and surface markers, the authors need to summarize or conclude the inference in affecting SARS-CoV2 infection as well as its implication in coinfection and comorbidities.
· Figure 3 could use some revision with better description of which markers are reported to change during the coinfection.
· Further, in case of asthma, which is usually treated by inhaled steroids, the authors should comment on the effect of medications on T cell subsets and if the reports of SARS-CoV2 co-infection in asthma discusses the medications used by the patients.
· Like influenza, SARS-CoV2 infection increases the risk of bacterial and fungal superinfections and coinfections. These increase the morbidity and mortality associated with infection. The authors need to discuss T cell subsets and outcomes in this context.
Minor concerns:
The review needs improvement in word usage. For example, “corrupt T cells” in case of HIV infection.
Page 1, line 36: “In diabetic patients, the clinical outcome is positively correlated with SARS-CoV-2 infection”. Do they mean diabetic patients have increased risk/prevalence of severe SARS-CoV2 infection?
Page 2, line 57: These cytokines help eliminate SARS-CoV2 infected cells. Unlike antibodies, they do not eliminate the virus directly.
Page 6, line 193: Viral RNA shedding is “prolonged” in coinfection.
Page 8, line 266: Allergens cause “airways” to become narrower due to inflammation.
Reviewer 3 Report
Overall, it is an informative review.
Perhaps the authors can describe very briefly the NKT cell responses which is especially important during the initial stage of infection where the cytokine storm occurs.
Figure 2 and Figure 3 are the same. The authors probably present a wrong figure 2 by mistake.
In section “2. T cell subsets and surface markers change…”: Perhaps the authors should briefly mention the function of each T cell subset as they are introduced. This will make it easier for non-experts to understand the consequence of any changes of the T cell subsets and surface markers.
Line 24: “since” should be “up to”
Line 71: Shouldn’t it be “Change of T cell subsets and surface markers …” instead of “T cell subsets and surface markers change…”
Line 55: Could the authors elaborate the meaning of the sentence “…, epitope pools contained higher CD4+ and CD8+ cells levels among convalescent SARS-CoV-2 patients”?
Line 77: “Meidleman” should be “Neidleman”
Line 79: “was contradict for”?
Reviewer 4 Report
In their paper, Wang and co. summarize current data relating to the influence of comorbidities on the outcome of SARS-CoV-2 infection, and how they affect the anti-viral T cell responses, focussing on conventional CD4 and CD8 responses. These comorbidities include HBV, HCV, HCV, TB and parasitic infections. They also include non-infectious comorbidities: diabetes, hypertension and asthma. The paper starts with a brief description of SARS-CoV-2 viral cycle and known alterations of T cell markers during SARS-CoV-2 infection.
The paper is well structured. We believe however that most statements lack precision and the message is frequently obscured by syntax errors and imperfect English. The paper would certainly benefit from being edited by a native English speaker. Alterations of immune T cell subsets in the different settings should be more carefully and comprehensively described
Major comments:
· Figure 2 is missing and replaced by a duplicate figure 3. Thus it cannot be evaluated.
· Figure 3 is unclear: activate/exhaust/exacerbate/corrupt/impair: what is modulated in each case? CD4 responses? SARS-Cov2 infection? coinfection? Global patient status?
Among unclear statements:
· Abstract, line 15: interrupt their clinical outcome: the meaning is unclear
· Introduction, lines 36-37: the clinical outcome is positively correlated…: the outcome of SARS or of diabetes?
· Line 55: epitope pools contained higher CD4….: does not seem appropriate.
· Lines 59-62: lower CD4 and CD8 frequencies in severe SARS-Co2: is this a cause or a consequence? This should be discussed in view of the questionable role of T cells in viral clearance.
· section.2: CD8 T cells: the message would be clearer by defining the markers of the different classical memory subsets at the beginning and then describing how they are altered in SARSCoV2 infection. An overall conclusion would be necessary for this section.
· Line90: CD27 by itself is not a senescence marker; in several instances, increased expression is mentioned when increased percentage of positive cells might be more appropriate (l.88, 90?)
· Section2: CD4 T cells: lines 110-111: imbalance: / abnormal secretion: it would be important to indicate clearly how a Th2 increase or decrease occurs in the different settings (mild or severe infection) or if this is not known.
· Line 117: cTFH17: define what these cells are.
· Lines 118-119: what sort of immunodeficiency?
· section 3.1 (164-179): coinfection is ambiguous: it should be clarified if this relates to SARS-CoV-2 infections occurring in HBV/HIV/HCV/TB carriers. In addition, it is never clear or discussed if the severity of clinical alterations is due to more severe SARS-CoV2 infection or aggravation of underlying disease.
· Line193: coinfection elongates viral RNA shedding: of which virus, HCV or SARSCov2?
· Section 3.2: 215-217: delay of CD4 and CD8 activation: is this general or specific for SARSCoV-specific cells? Or specific for anti-Mtb cells ?
· Line 220: What are Latent-SARS-Cov-2 patients?
· Section 3.3: L.227-230: precise which parasitic infections have been studied.
· Section 4.2: line 266: in “cause the lungs to become narrower” : shall we understand bronchial tubes instead of lungs ?
· Section 4.3: line 312: fetal ?
Round 2
Reviewer 4 Report
The manuscript has been very significantly improved.